# Passive tracer advection in the equatorial Pacific region: statistics, correlations, and a model of fractional Brownian motion

Imre M.  Jánosi[1,2], Amin Padash[1], Jason A. C. Gallas[1,3], and Holger Kantz[1]

[1]Max Planck Institute for the Physics of Complex Systems, Nöthnitzer Str. 38, 01187 Dresden, Germany

[2]Department of Water and Environmental Policy, Faculty of Water Sciences, University of Public Service, Ludovika tér 2, H-1083 Budapest, Hungary

[3]Instituto de Altos Estudos da Paraíba, Rua Silvino Lopes 419-2502, 58039-190 João Pessoa, Brazil

**Correspondence:** Imre M. Jánosi (janosi.imre.miklos@uni-nke.hu)

**Abstract.**

Evaluating passive tracer advection is a common tool to study flow structures, particularly Lagrangian trajectories ranging from molecular scales up to the atmosphere and oceans. Here we report on numerical experiments in the region of the tropical Pacific ($20°$S - $20°$N), where 6600 tracer parcels are advected from a regular initial configuration (along a meridional line at $110°$W between $15°$S and $15°$N) during periods of one year, 25 years altogether. We exploit AVISO surface flow fields and solve the kinematic equation for passive tracer movement in the 2D advection tests. We demonstrate that the strength of the advection defined by mean monthly westward displacements of the tracer clouds exhibit surprisingly large inter- and intra-annular variabilities. Furthermore an analysis of cross-correlations between advection strength and El-Niño and Southern Oscillation Indices (SOI) reveal a significant anti-correlation between advection intensity and ONI (Oceanic Niño Index) and a weaker positive correlation with SOI, both with a time lag of about 3 months (the two indices are strongly anticorrelated near real-time). The statistical properties of advection ( time dependent mean squared displacement and first passage time distribution) suggest that the westward moving tracers can be mapped into a simple 1D stochastic process, namely fractional Brownian motion. We fit the model parameters and show by numerical simulations of the fractional Brownian motion model that it is able to well reproduce the observed statistical properties of the tracers' trajectories. We argue that a traditional explanation based on the superposition of ballistic drift and a diffusion term yields different statistics and is incompatible with our observations.

## 1   Introduction

Studies of regular or chaotic advection of various tracer particles are a growing area of research thanks to measurements of increasing sampling densities and numerical models of ever-better resolutions (Aref et al., 2017). Phenomena related to advection cover a very wide range from molecular scales to geophysical flow fields in the atmosphere and oceans. The starting point of the subject is certainly the simplest case, passive tracer advection (Aref, 1984; Ottino, 1990). In a fluid treated as a continuum, one can conceptually mark a particle (microscopic parcel) which moves passively with the flow velocity **v** obeying

the simple kinematic equation

$$\frac{\mathrm{d}\mathbf{r}(t)}{\mathrm{d}t} = \mathbf{v}\left(\mathbf{r}(t)\right) \ , \tag{1}$$

from an initial position $\mathbf{r}(0) = \mathbf{r_0}$. The resulting Lagrangian trajectories $\mathbf{r}(t)$ permit an insight into the details of flow structures and mixing. A fundamental finding is that very simple time-periodic flow fields may advect tracers chaotically indicated by an exponential divergence from close initial positions. In most cases, it is possible to reconstruct invariant sets of fractal nature in the phase space related to chaotic advection (see e.g. Péntek et al., 1995; Cartwright et al., 1999; Tél et al., 2000; Speetjens et al., 2021). Recent important applications are connected to the exploration of Lagrangian coherent structures (Haller, 2015; Hadjighasem et al., 2017; Haller et al., 2018; Beron-Vera et al., 2018; Callies, 2021; Haller et al., 2021), mostly in the context of ocean flows.

Studies of advection and material transport in the oceans have a growing importance partly because of catastrophes like the Deepwater Horizon explosion (Gulf of Mexico, 2010), see e.g., Olascoaga and Haller (2012), or the nuclear disaster in Fukushima (Japan, 2011), see e.g., Prants et al. (2017). Besides these extreme events, "traditional" interest also has been growing in transport of planktons (Károlyi et al., 2000; Huhn et al., 2012; Hernández-Carrasco et al., 2020; Villa Martín et al., 2020), heat (Webb, 2018; Adams et al., 2000; Ruggieri et al., 2020), salt (Delcroix and Picaut, 1998; Sanchez-Rios et al., 2020; Paul et al., 2021), oxygen (Rudnickas et al., 2019; Audrey et al., 2020) and other chemicals (Behrens et al., 2020; Jones et al., 2020). Another classical list of works aims to determine mixing rates by determining suface eddy diffusivities (see e.g., Abernathey and Marshall, 2013, and references therein).

Considering ocean surface current systems, the equatorial Pacific is certainly a key region from many respects (see e.g., Reverdin et al., 1994; Yu and McPhaden, 1999; Grodsky and Carton, 2001; Chepurin and Carton, 2002; Capotondi et al., 2005; Izumo, 2005; Kessler, 2006; Bulgin et al., 2020; Wang et al., 2020; Power et al., 2021). The El Niño - Southern Oscillation (ENSO) has a near global impact on weather, agricultural yields, air quality and even landslides (Grove, 1998; Collins et al., 2010; Chang et al., 2016; Luo et al., 2018; Timmermann et al., 2018; Emberson et al., 2021; Power et al., 2021). By means of recent observations (mostly Argo buoys) and high resolution numerical ocean models, it became clear that the tropical Pacific region obeys a complicated 3D structure. Besides the surface currents, determining features are the equatorial upwelling zone resulting in poleward Ekman transport, the equatorial undercurrent (an energetic eastward flow at the depth of the pycnocline $\sim$50-200 m), and the shallow meridional overturning circulation both north and south of the equator.

Here we focus on the tropical Pacific region and compute Lagrangian trajectories by passive tracer advection approach using Eq. (1). The velocity fields are obtained from AVISO altimetry data (Aviso; Taburet et al., 2019) which has a higher spatial resolution ($0.25° \times 0.25°$) than the global ocean numerical models (usually with $1° \times 1°$), whereas in regional ocean models the model resolution can be as fine as 3-9 km (Nefzi et al., 2014). We will demonstrate that the overall behavior of tracers is an anomalous diffusion process which is slower than a pure drift but faster than simple diffusion. The novel observation here is that the Hurst exponent of this anomalous diffusion process is a constant value for a large range of spatial and temporal scales, from 1 to 5000 km and from 2 to 365 days. This means that the collective effects of the spatial and temporal fluctuations of the velocity field which advects the particles has some self-similar structure which gives rise to a rather uniform time evolution

*on average over several years.* Instead of a constant Hurst exponent over such large ranges of spatial and temporal scales, one might expect some cross-over effects, e.g., from (anomalous) diffusion on small scales to ballistic, drift dominated transport on larger scales. The absence of such cross-over shows that the irregularities of the velocity fields in space and time do not average out even on large scales and might reflect some kind of scale invariance of the 2D turbulent motion of these hydrodynamic flows. This provides insight into the statistics of oceanic turbulence and might be of fundamental physical interest.

An alternative description would be a model with an explicit ballistic drift term plus normal diffusion. The drift term would then describe the time dependence of the mean value of the zonal particle positions, and the normal diffusion term would explain why the variance of this distribution grows in time. Our analysis clearly shows that the statistical properties of such a quite plausible model are incompatible with the observations based on the numerical tracer advection.

In the next Section we summarize the data sources and methodologies of the statistical analysis. The Section "Results and discussions" gives an overview regarding the large variability we obtained in the strength of the advection, and cross-correlation properties with the El Niño, and composite Southern Oscillation indices. The trajectories indicate a strong westward drift and relatively weak dispersion in the meridional directions, which provides a mapping to a one-dimensional stochastic random walk model. Statistical analysis of the mean squared displacement in zonal direction and of first passage times suggest that an appropriate such model is the fractional Brownian motion (fBm) with some small deterministic westward drift. We fit the parameters of this model and validate it by numerical simulations. We demonstrate that a deterministic drift term provides negligible improvement so that we end up in a 2-parameter fBm model.

## 2 Data and methods

Geostrophic surface velocity fields were obtained from the AVISO data bank (Aviso; Taburet et al., 2019). Geostrophic balance does not hold at the equator, but the altimetry data can still be used to infer velocities there, with somewhat lower accuracy. The AVISO data processing algorithm implemented the method of Lagerloef et al. (1999) between $\pm 5°$ latitudes (Aviso; Taburet et al., 2019). We used the direct zonal and meridional velocity components (*ugos* and *vgos*, in units of m/s). The spatial resolution of daily global records is $0.25° \times 0.25°$ ($1440 \times 720$ grid-cells), land areas are masked. The temporal resolution is 1 day between the period 1 Jan 1993 - 23 Oct 2018, however we cut the last truncated year for the proper comparison of annual results. Offline passive tracer advection were estimated by bilinear interpolation of velocity values, and solving Eq. (1) by a fourth order Runge-Kutta method with a time step of 5 min. The position of advected parcels was recorded in every 12 hour throughout a given year (365 days). On January 1st of each year, 6600 tracers were started from a meridional line at the longitude 110°W, between latitudes 15°S - 15°N (approximately 500 m spacing). All calculations were performed using the package `Ocean Parcels` (Lange and van Sebille, 2017; Delandmeter and van Sebille, 2019) in a Python environment (version 3.6) with the standard `Numpy` (Harris et al., 2020) and `Scipy` (Virtanen et al., 2020) libraries, maps drawn by the `Cartopy` module (Met Office, 2010 - 2015). As representative examples, Fig. 1 illustrates two consecutive years, 1997 and 1998. 1997 was a very strong El Niño year, and Fig. 1a demonstrates clearly the well known effect of weakened (easterly) trade winds in such years.

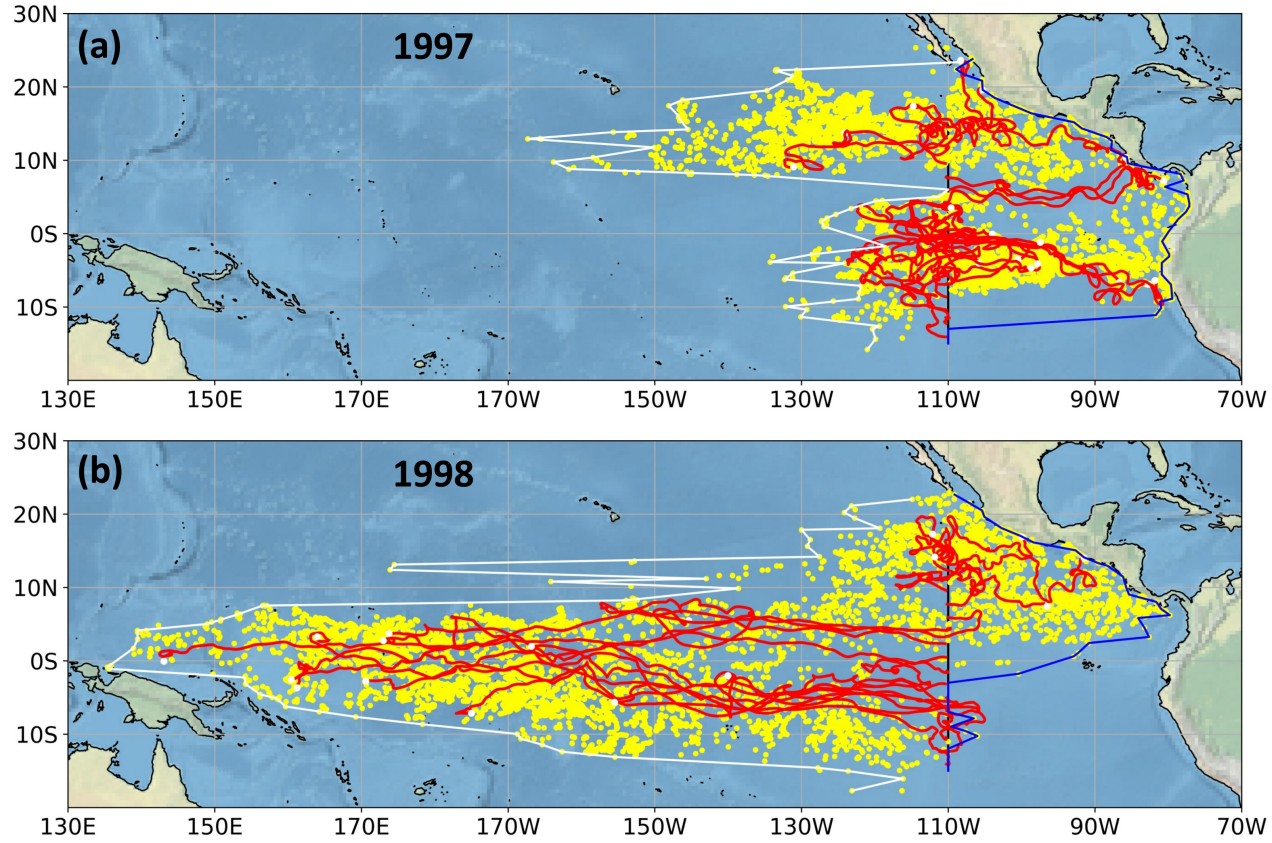

**Figure 1.** The final position of passively advected parcels (yellow dots) after 365 days starting from 1st of January in (a) 1997 (a particularly strong El Niño year) and (b) 1998. The initial configuration of 6600 tracers is a meridional line at the longitude 110°W, between latitudes 15°S - 15°N (black vertical line on the maps). The white/blue lines indicate the western/eastern envelopes of the tracer clouds after one year of advection, red lines exhibit representative trajectories. The flow fields are from the AVISO data bank (Aviso; Taburet et al., 2019).

Similar numerical experiments are reported by Webb (2018), where hot parcels (T > 27°C) are advected in two years (1981 and 1982) from 24 June to the end of the year from an initial meridional line at the middle of Pacific, north to the equator (see Figs. 26 and 27 in Webb, 2018). The high variability of the North Equatorial Counter Current (NECC) is nicely demonstrated.

We have no access to the vertical velocity field, and no model for the advection of particles in 3D with the ocean surface as a boundary. Clearly, real transport is not restricted to the topmost horizontal layer of the ocean, but it is a useful approximation and

also a standard approach to ignore the third dimension in such transport studies. We assume that our numerically advected tracer paths are sufficiently realistic compared to, e.g., floaters released to the ocean. Note, however, that a direct comparison of our analysis with trajectories of drifting surface buoys or satellite altimetry is not straightforward, because our initial configuration (parcels arranged along a meridional line) is rather specific for each year. Furthermore surface buoys are directly affected by

surface wind shear, thus their drifts not always reflect the displacement of underlying water parcels (Reverdin et al., 1994; Grodsky et al., 2011).

From the advection data for each year, we adopted two definitions of an "advection index" (AdI) to characterize the westward drift strength. A rather large fraction of parcels travel to the east from the initial position and remain trapped in the eastern equatorial Pacific (see Fig. 1). However, in this work we do not consider their complicated behavior, rather focus on the westward moving tracers. Monthly advection indices $AdI_1$ and $AdI_2$ are defined by the ensemble mean westward distance and total mean trajectory length from the positions at the end of the previous month. We will see that there is no difference between the values of both indices, demonstrating that westward drift absolutely dominates.

In order to connect the AdI with the well-known large scale climate patterns, we determined cross-correlations with two standard indices, SOI (Southern Oscillation Index) and ONI (Oceanic Niño Index). (Actually we checked other distant index values, like Arctic Oscillations, Antarctic Oscillations, etc., but we obtained meaningful correlations only with the two "local" diagnostic quantities.) The standardized SOI is obtained from the monthly or seasonal differences in the air pressure between Tahiti and Darwin, Australia (Trenberth and for Atmospheric Research Staff , Eds). ONI is almost the same as the "traditional" Niño 3.4 index, being based on mean sea surface temperature anomalies over the area of [5°S - 5°N, 170°W - 120°W]. ONI is the operational definition used by NOAA to define an El Niño or La Niña event.

Normalized cross-correlations between two signals $x(t)$ and $y(t)$ are determined as usual:

$$X(\tau) = \frac{\langle [x(t) - \overline{x}][y(t+\tau) - \overline{y}] \rangle}{\sigma_x \sigma_y} \ , \tag{2}$$

where the time lag $\tau$ represents a temporal shift of $\pm\tau$ months between the two time series, overbar denotes temporal mean, and $\sigma$ is the standard deviation of the given time series. Note that according to the above definition (realized in the `scipy.signal.correlate()` routine), the second signal $y(t)$ moves along the time axis in both positive and negative directions with respect to $x(t)$. Thus a *positive* lag correlation means that $y(t)$ leads $x(t)$, and vice versa.

The confidence interval (at a level of 99%) for cross-correlations is obtained in the following way. It is well known that the presence of serial or higher order auto-correlations in $x(t)$ and $y(t)$ can yield spurious cross-correlations (e.g., Zwiers, 1990; Ebisuzaki, 1997; Massah and Kantz, 2016). Following the proposal of e.g., Ebisuzaki (1997), we computed 10,000 surrogate data sets $\{x_i'(t)\}$ for $x(t)$ by the method of iterative amplitude adjusted Fourier transform (IAAFT) (Kantz and Schreiber, 1997; Hegger et al., 1999; Schreiber and Schmitz, 2000; Lancaster et al., 2018). Such surrogates reproduce the amplitude distribution and power spectrum of the source record $x(t)$. We used 100 iterations in a given run to build the ensemble $\{x_i'(t)\}$. Then we calculated $X(\tau)$ by Eq. (2) for each $[x_i'(t), y(t)]$ pairs of signals. After sorting cross-correlation values for each time lag $\tau$ in an increasing order, the confidence interval of 99% were obtained by the precentile range [0.05, 99.5] (from an ordered test ensemble of 10,000 members we dropped the lowest 50 and highest 50 values). Note that the method of surrogate data is a modification of classical bootstrapping, where both the marginal distribution of the data and their serial correlations are conserved when creating the otherwise random samples.

The statistical analysis of advection is based on two standard measures, the First Passage Time (FPT) (Metzler et al., 2014b) and Mean Square Displacement (MSD) (Jánosi et al., 2010; Haszpra et al., 2012; Kepten et al., 2015). Both measures have a

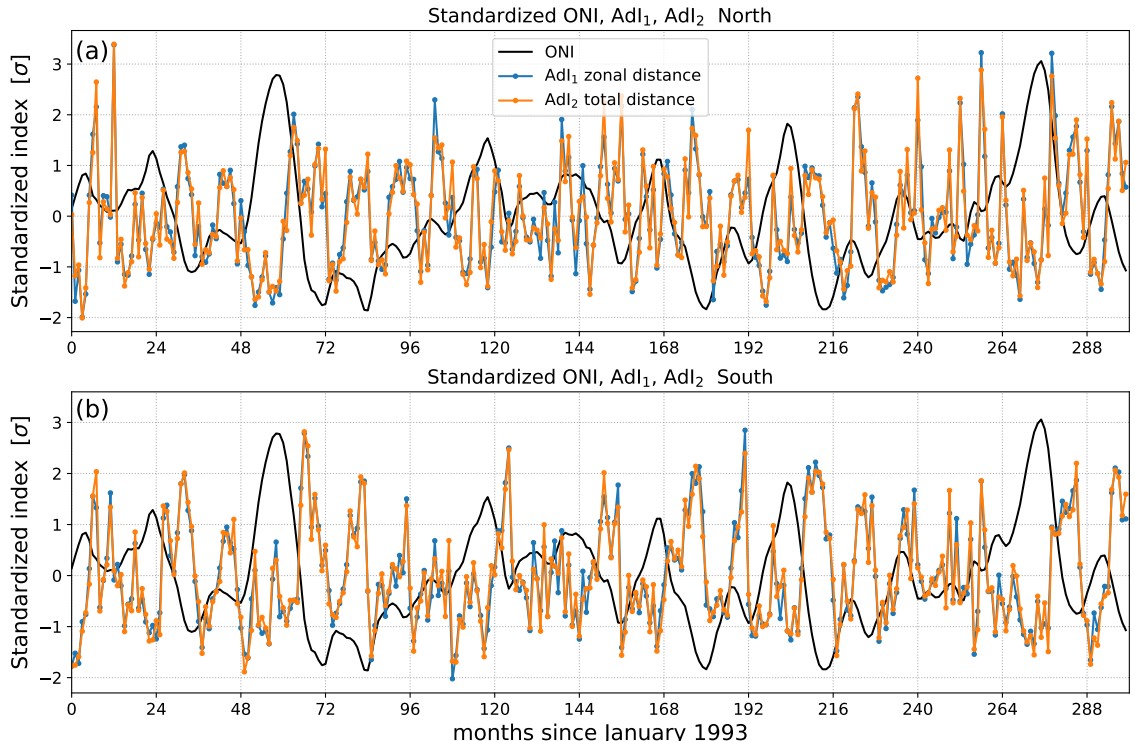

**Figure 2.** Standardized advection indices $AdI_1$ and $AdI_2$ determined as ensemble averages of the monthly westward displacements. The first is based on the monthly zonal distance from the position at the end of the previous month, the second is the total trajectory length in a month (meridional components are also taken into account). For a comparison of variabilities, the ONI (Oceanic Niño Index) is also plotted by black lines. **(a)** northern section; **(b)** southern section.

transparent definition: in our setup, FPT is the time when a given longitude is crossed by a parcel, and MSD is calculated as the time evolution of the squared distances from the initial location. In the second case, a refined statistics is also adopted by computing the Time Averaged Mean Square Displacement (TAMSD) (Lubelski et al., 2008; Kepten et al., 2015; Sikora et al., 2017; Maraj et al., 2021). In this case, a whole trajectory of a parcel is evaluated by introducing a time window parameter $w$, which changes from 12 h to 315 days ($2 \times 315$ time steps in the record). This time window moves along the trajectory step by step, and a temporal mean is computed for the squared displacements in a window $w$. This procedure is computationally rather demanding, but it smooths out e.g., seasonal and intraseasonal fluctuations.

 ## 3   Results and discussions

### 3.1   Advection index

As discussed in the previous Section, in order to determine the strength and variability of the advection, we use two (monthly) advection indices $\text{AdI}_1$ and $\text{AdI}_2$. The subsequent analysis distinguishes two subsets of tracers, based entirely on their initial positions: parcels north/south to the geographic equator are separated into northern/southern sections, because *a priori* it is not

clear whether these two advection processes have the same statistical properties. The reason of this simplistic subset definition is that hemispherical surface winds and surface ocean currents in a given short period are determined not by the geographic zero latitude but rather by the "thermal" or "heat" equator, which actually has a strong coincidence with the Intertropical Convergence Zone (ITCZ) (Amador et al., 2006a; Kessler, 2006b). The location of the ITCZ or, equivalently, the thermal equator is not simply zonal, and has an annual cycle following the changes of the incoming shortwave solar radiation flux.

The mean position of it is around $5°N$, because the ice mass of the Antartic and the lower fraction of land on the southern hemisphere result in a lower mean temperature south of the geographic equator. This mean location is clearly visualized by maps of annual precipitation in the region (see e.g., Fig. 7. in Amador et al., 2006a), or by the long time mean position of the NECC (see e.g., Fig. 1 in Hsin and Qiu, 2012) or (Figs. 1 and 3 in Wijaya and Hisaki, 2021). The dynamic relocation of ITCZ in a year has the consequence that trade winds and ocean surface currents are also changing, thus advected parcel trajectories

often cross the geographic equator and the mean thermal equator. There are some years (usually of weaker advection strength) when the separation of the northern and southern drifting parcels are clear, and the presence of NECC is clear (see Fig. 1a). In years of usually strong advection, such separation is not so clear (see Fig. 1b).

We note here that the instantaneous (daily) location of the ITCZ line is a delicate question, that is why ITCZ is usually illustrated by long time mean values in the literature. We could not figure out an adequate method to separate parcels along the

two sides of the ever changing ITCZ line.

Fig. 2 exhibits the advection indices together with the ONI (Oceanic Niño Index). The definitions of $\text{AdI}_1$ and $\text{AdI}_2$ result in almost identical standardized monthly mean values indicating that pure zonal drift dominates for westward moving parcels. (Standardization is obtained in the usual way by removing long time mean values and normalizing by the standard deviation.) A simple visual inspection is enough to detect substantial differences between the northern and southern subset of parcels

(compare Fig. 2a and Fig. 2b). Indeed, when we determine cross-correlations [Eq. (2)] for the northern and southern advection indices, there are some remarkable features, see Fig. 3.

The first remarkable aspect in Fig. 3 is that the real-time positive correlation is rather moderate ($X(0) = 0.55$), thus the northern and southern Pacific currents are not strongly coupled. The second aspect is the presence of a weaker but statistically significant (at the level of 99%) anti-correlation with a time lag of 7 months ($X(\tau = 7) = -0.31$). Similar anti-correlations

is usually explained by the seasons on the two hemispheres appearing in a counter-phase. For this reason, practically each climatology considers monthly mean values and anomalies automatically assuming a time lag of 6 months. Note that in Fig. 3, not only the minimum place but the fainting local maxima of positive correlations do not precisely follow the periodicity of

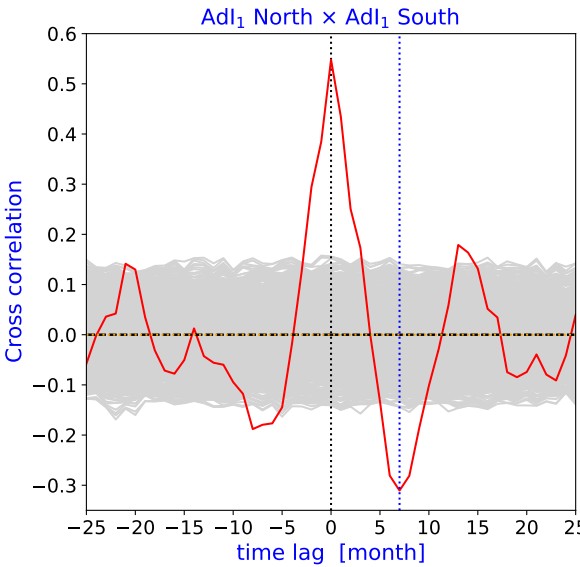

**Figure 3.** Cross-correlation between the monthly advection index $AdI_1$ for the northern and southern subsets of advected parcels (red). Blue/black dashed lines indicate the time lags where the cross-correlation has the absolute minimum/maximum. The gray band indicates the 99% confidence interval as described in Section 2.

one calendar year. While seasonality in the weather is not essential in the tropical band, the annual cycle of the location of the thermal equator follows the changes of insolation.

## 3.2 Cross-correlations with climate indicators ONI and SOI

Figs. 4 and 5 illustrate cross-correlations Eq. (2) between two monthly climate indicators ONI and SOI and the advection index $AdI_1$ separately for the two hemispheres, and also for the joint $AdI_1$ incorporating monthly zonal distances travelled by all the westward drifting parcels. As for ONI, the main feature is an absolute negative minimum at a time lag of 3 months, SOI has a positive correlation at the same time lag. Furthermore a weaker but significant positive correlation with ONI and negative correlation with SOI (at least in the Southern basin) appear with a time lag of -7 months. All these are fully consistent with the well-known strong real-time anti-correlation between ONI and SOI illustrated in Fig. 6 (we show here this figure to emphasize that the apparent positive peaks at -2 and +2 years are not significant at the 99% confidence level).

Experts can skip this paragraph, it is intended to those readers who have primary interest in fractional Brownian motion, without a background of environmental flows. El Niño was named first in print in the context of climate around South America as far back as 1893. The Southern Oscillation earned its name somewhat later in 1924. The observations and detections of both climate phenomena can be traced back to colonial times, as summarized by Grove and Adamson (2018). Southern Oscillation was a key research focus for Walker during the late stage of his career, he was the first describing the quasiperiodic behaviour of the pressure contrast with a mean period of three and half years (Grove and Adamson, 2018). The research on oceanic El

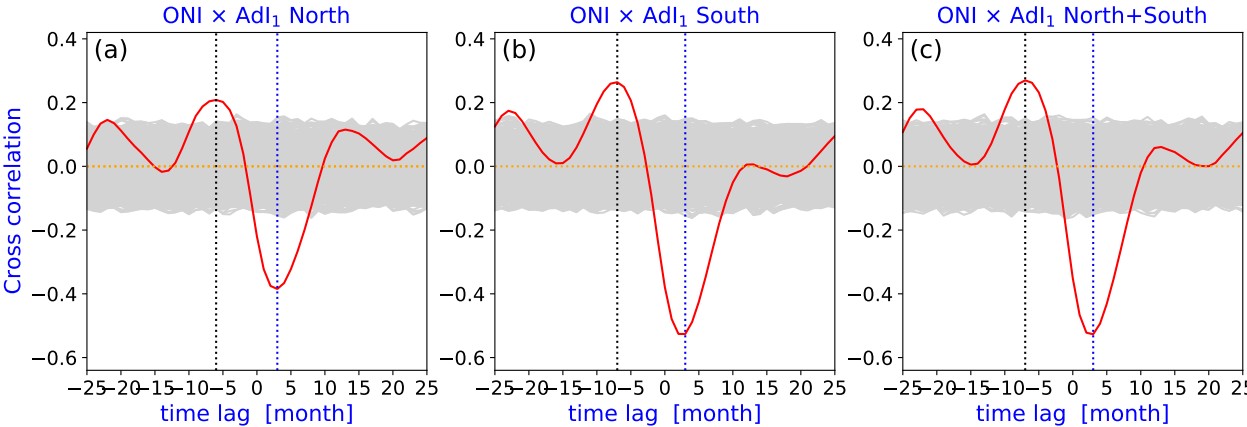

**Figure 4.** Cross-correlation between ONI and the monthly advection index $AdI_1$ for the northern and southern subsets of advected parcels (red). Blue/black dashed lines indicate the time lags where the cross-correlation has the absolute minimum/maximum. The gray band indicates the 99% confidence interval as described in Section 2. (a) $AdI_1$ for the northern subset (see Fig. 2a). (b) $AdI_1$ for the southern subset (see Fig. 2b). (c) Cross-correlation between ONI and the monthly advection index $AdI_1$ for all the westward moving parcels. The strongest anti-correlation at +3 months in each case means that advection strength $AdI_1$ leads ONI: a marked weakening of advection is followed by a warming SST signal with a delay of 3 months, and vice versa.)

Niño and atmospheric Southern Oscillation run parallel but essentially independent for decades, until the landmark paper that finally linked El Niño and the Southern Oscillation, published by Bjerknes in 1969. The merging of the two names as El Niño - Southern Oscillations has its origin somewhat later, in a paper by Rasmusson and Carpenter in 1982 (Grove and Adamson, 2018).

Since the two quasiperiodic oscillations ONI and SOI are so strongly anticorrelated in near real-time (Fig. 6), the behavior of the advection index follows what is expected. The interpretation of the time lag 3 months is not so trivial. Obviously, correlation is not equivalent to a causal link, the only information which can be extracted with high confidence is a temporal sequence of events. The time lag of 3 month suggests that the ONI index rises (the Niño 3.4 area warms up), and parallel with it the SOI index decreases (air pressure anomaly between Tahiti and Darwin drops), three months later than the advection strength decreases significantly. In the opposite direction the behavior is analogue, when ONI drops and SOI increases, westward advection is accelerated three month earlier. This rather long delay is somewhat perplexing, because it is known that the characteristic response time of the ocean surface currents for zonal wind bursts (lasting typically 5-15 days) is a few days, at most. This is observed around the Antarctic Circumpolar Current (Webb and de Cuevas, 2007), and also at the western sector of the equatorial Pacific (McPhaden et al., 1992; Delcroix et al., 1993; Richardson et al., 1999). It is well known that during an El Niño event the prevailing trade winds and the surface westward currents suffer from a temporary reversal in the western part of Pacific basin. Our analysis suggests that this reversal blocking advection propagates toward the eastern basin for about three months.

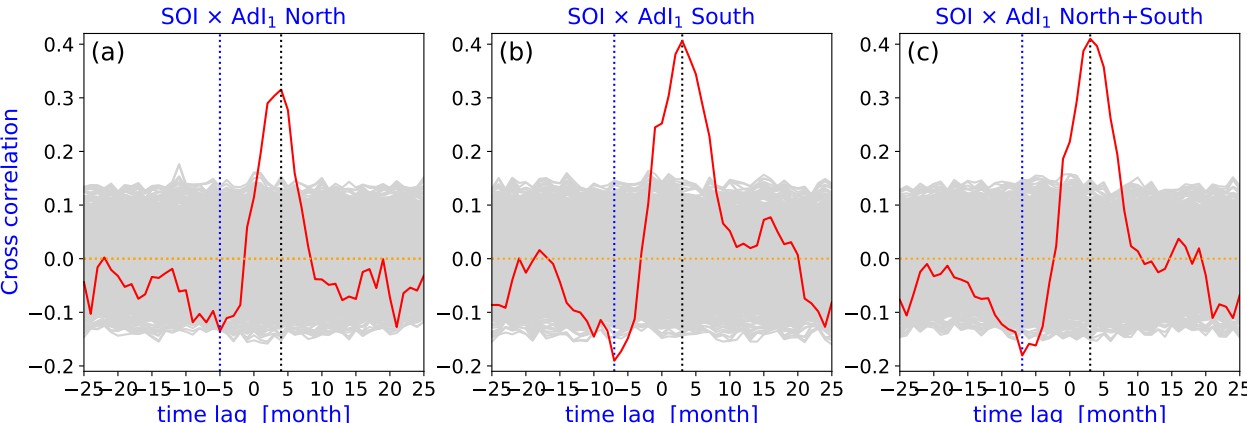

**Figure 5.** Cross-correlation between SOI and the monthly advection index $AdI_1$ for the northern and southern subsets of advected parcels (red). Blue/black dashed lines indicate the time lags where the cross-correlation has the absolute minimum/maximum. The gray band indicates the 99% confidence interval as described in Section 2. (a) $AdI_1$ for the northern subset (see Fig. 2a). (b) $AdI_1$ for the southern subset (see Fig. 2b). (c) Cross-correlation between SOI and the joint monthly advection index $AdI_1$ for all the westward moving parcels. The strongest correlation at +3 months in each case means that advection strength $AdI_1$ leads SOI: a strengthening of advection is followed by an increasing pressure anomaly (above-normal air pressure at Tahiti and below-normal air pressure at Darwin, Australia) with a delay of 3 months, and vice versa.

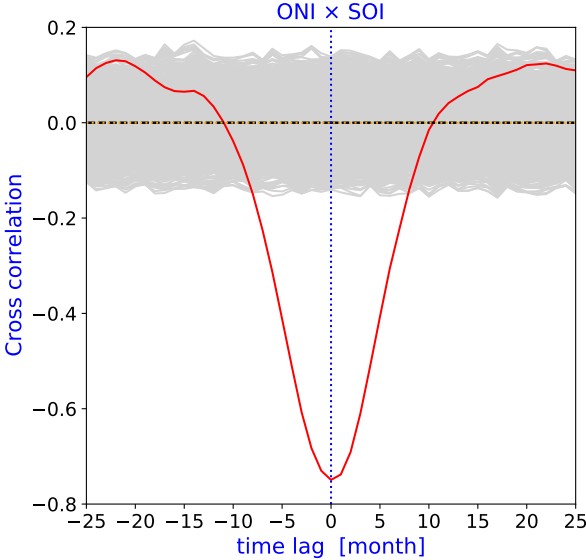

**Figure 6.** Cross-correlation between the monthly index values of ONI and SOI (red). Blue dashed lines indicates the zero time lag where the cross-correlation has the absolute minimum.The gray band indicates the 99% confidence interval as described in Section 2.

We already mentioned that the tropical Pacific currents have a pretty well-known complex 3D stucture. Capotondi et al. (2005) and Izumo (2005) determined lag correlations between various branches (equatorial undercurrent and shallow meridional overturning cells) and the spatial mean of sea surface temperature over the Nino3.5 area. The high resolution 3D ocean models found significant cross-correlations of various time lags on the scale of months. Izumo (2005) obtained anti-correlations between the SST signal and surface divergence (poleward Ekman transport) with a lag of 5.5 months, equatorial upwelling ($\tau = 5$ months), equatorial undercurrent ($\tau = 3$ months), and pycnocline convergence ($\tau = 0.5$ month) suggesting that the changes in the 3D current structure lead the appearance of an El Niño signal. An interesting result by Capotondi et al. (2005) relates equatorward mass convergence along the pycnocline and equatorial upwelling with SST. The strongest correlation is found when SST leads meridional mass convergence by 2 months, a result that seems to be in disagreement with the view that changes in the strength of the subtropical-tropical cells (STCs) cause the SST changes. Since we do not have access to the 3D flow structure, we cannot relate our finding (SST lags behind advection strength changes by ∼3 months on the very surface) with the results of Capotondi et al. (2005). They notice, however that "phase relationships among the different STC components can be largely affected by the zonal averaging procedure because of the continually evolving nature of the STCs".

### 3.3 Statistical properties of westward advection

From Fig. 1 it becomes evident that the tracers perform a highly complex motion which contains many aspects of stochasticity. Therefore, it is natural to analyse the tracer trajectories in terms of a diffusive process. We do so by focusing on three essential characteristics which allows us to better understand which stochastic process is best suited to explain the observed phenomena.

Fistly, we determine the *mean squared displacement* (MSD) as a function of time which is defined as the average over many trajectories (i.e., an ensemble average) of the square of the distance between the initial point and of the endpoint at time $t$ of each trajectory, in formula:

$$\mathrm{MSD}(t) := \langle (x(t) - x_0)^2 \rangle , \tag{3}$$

where $\langle \ldots \rangle$ denotes the average over all the trajectories of westward tendencies. For a classical diffusion process, it is known that $\mathrm{MSD}(t) \propto t$, while for ballistic motion (i.e., a purely deterministic drift), $\mathrm{MSD}(t) \propto t^2$. Empirically, we find $\mathrm{MSD}(t) \propto t^{2H}$ with $H \approx 0.9$ (see Fig. 7), i.e., a faster than linear growth in time, but still less fast than ballistic motion. Such behavior is called *super-diffusion* (see e.g., Bourgoin, 2015, and references therein), and in models it is caused by certain properties of the noise. For stationary increments, i.e., for a time-independent distribution of the noise, super-diffusion can be either caused by long range temporal correlations, when the noise is not white but its auto-correlation function decays with the power law $c(\tau) \propto \tau^{-2H-1}$ and hence there is no finite correlation time. Alternatively, the noise can stem from a distribution with fat tails, so that it does not have a finite second moment. In this case, rare but huge jumps of the diffusive path let particles disperse much faster than diffusively. Any kind of superposition of these two mechanisms is also possible, so that there are many different models which behave super-diffusively. We will employ additional analysis in order to identify the origin of this anomalous diffusive behavior.

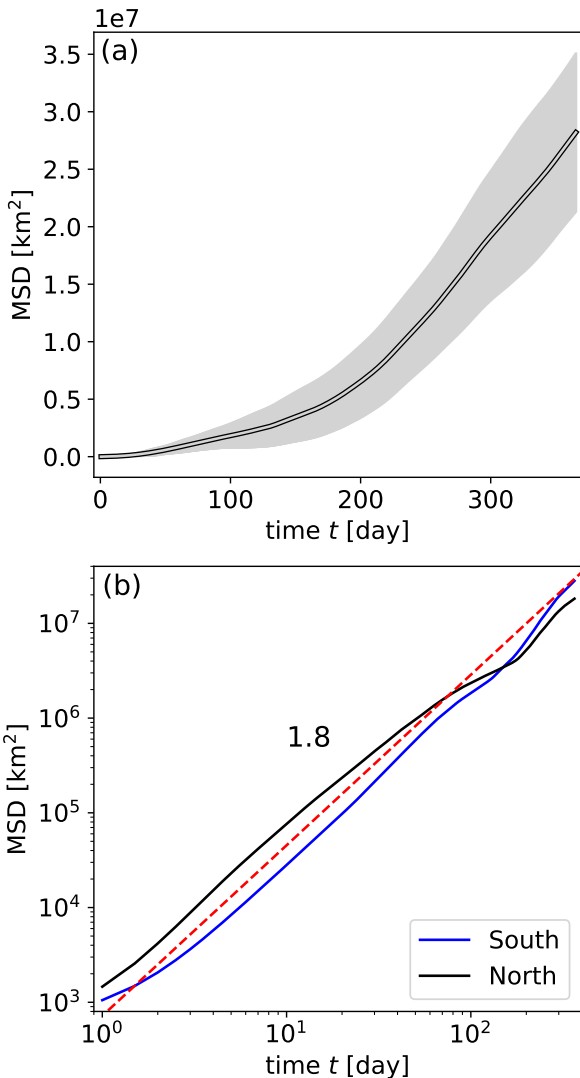

**Figure 7.** Mean Squared Displacement (MSD) for the advected parcels in units of km$^2$. The spherical geometry is fully taken into account at calculating distances from the initial positions. (a) MSD for the northern subset (lin-lin scales), the error band reflects the standard deviation obtained from year by year statistics. (b) MSD for both the northern (black) and southern (blue) subsets of parcels on a double logarithmic scale. The red dashed line illustrates power law with an exponent value around 1.8.

One such additional statistics is the time averaged mean squared displacement TAMSD, see Fig. 8. Empirically, it displays a power law scaling with the same exponent $\approx 1.8$ as the MSD. This is a signature of ergodicity (Metzler et al., 2014a), which restricts the choice of models further.

As third statistics, we study the first passage time distribution. This is the distribution of times needed by the individual trajectories to pass through some pre-defined longitude circle. Since all particles start at $110^oW$, this study considers the zonal

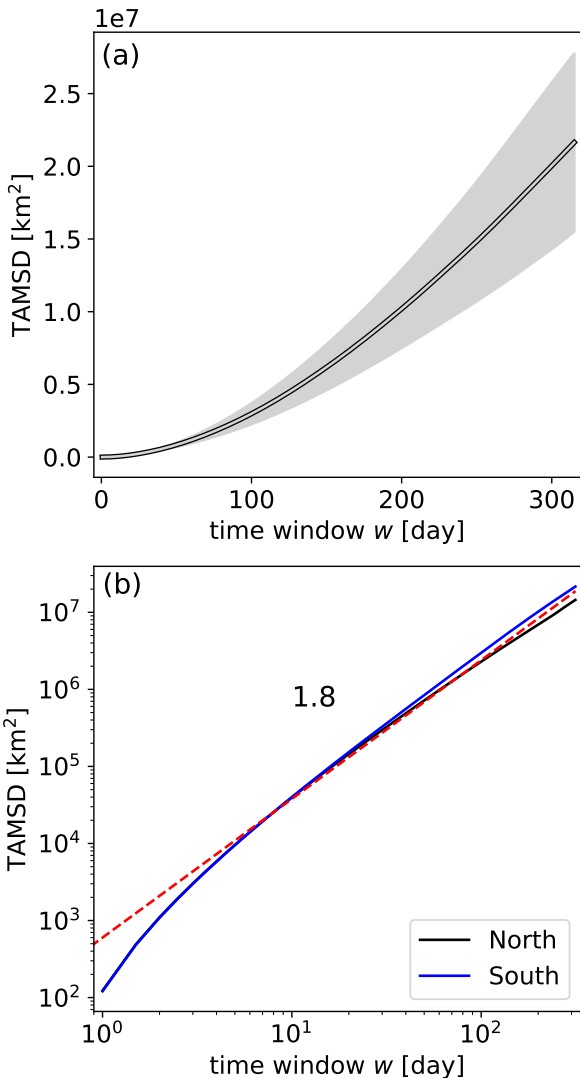

**Figure 8.** Time Averaged Mean Squared Displacement (TAMSD) for the advected parcels in units of km². The spherical geometry is fully taken into account at calculating distances in the moving time windows of size $w$ (a) TAMSD for the northern subset (lin-lin scales), the error band reflects the standard deviation obtained from year by year statistics. (b) TAMSD for both the northern (black) and southern (blue) subsets of parcels on a double logarithmic scale. The red dashed line illustrates power law with an exponent value around 1.8.

motion of the tracers. We study 4 different positions of the line for passage, namely those which are $1^o$, $2^o$, $4^o$, and $8^o$ west of the starting position. The corresponding distributions are shown in Fig. 9. The universal property of these distributions is a steep increase after some minimum time and a power-law decrease, $t^{-\alpha}$. The power $\alpha \approx 1.1$ is independent of the line to be passed and it is a further characteristic of the underlying stochastic process.

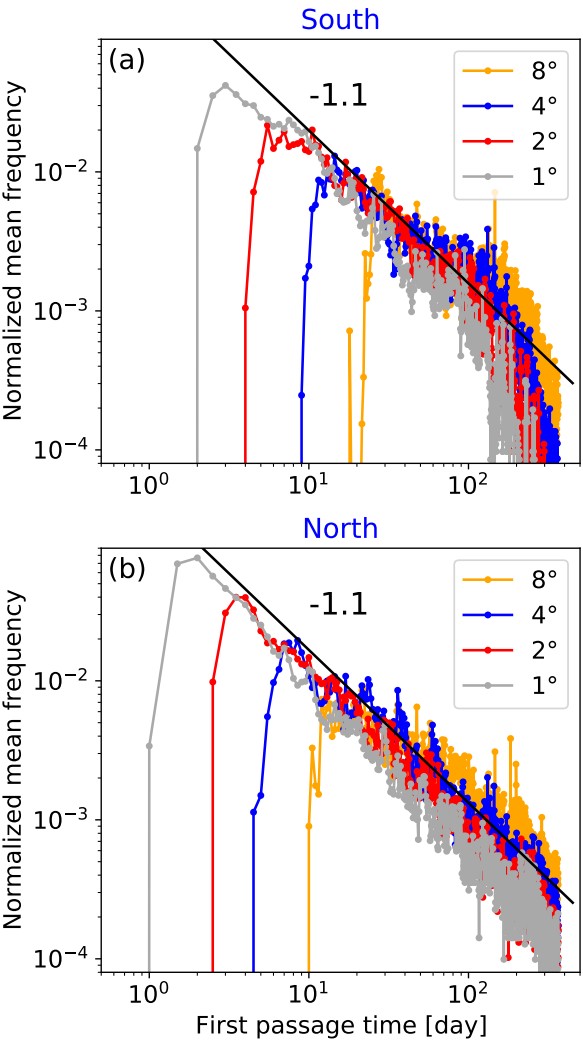

**Figure 9.** Normalized empirical frequency distribution of first passage time (FPT) for the (a) southern, and (b) northern subset of advected parcels on a double logarithmic scale. Four different boundaries are shown at 1, 2, 4 and 8 degrees west from the initial positions of tracers. The straight black line parallel with the envelopes illustrate a power law with an exponent value around -1.1 .

We evaluated the mean squared displacement of the particles up to distances of 10,000 km, and on times up to one year (Fig. 7), where the fractional Brownian motion model reproduces well the observed anomalous diffusion on the whole range of scales (see below). However, the nontrivial scaling of the MSD alone is not enough to identify fractional Brownian motion as the correct model among several other anomalous diffusion processes, so that we performed in addition the study of the first passage times. In this specific aspect, indeed, we restricted ourselves to distances of at most 900 km and 100 days. The reason for this is that in many years (one is shown in Fig. 1a), a large fraction of the particles does not cross the 8-degree-distance

from the place where they have been released. This means that the number of particles which contribute to the first passage time distribution becomes the smaller the larger the distance, and already for 16 degrees we no longer have a good enough statistical sampling of the probability density.

## 3.4 Fractional Brownian motion model

We can explain both behaviours of the tracers, that of the first passage time and that of the MSD, by the model of fractional Brownian motion (fBm). This is the path generated by accumulating (integrating) fractional Gaussian noises over time, with 2 free parameters: One is the Hurst exponent $H$, the other is a generalized diffusion coefficient $k_H$ which determines the amplitude of the noise. For this model it has been shown that the MSD scales like $t^{2H}$ (Metzler and Klafter, 2000, 2004), i.e. the scaling of the MSD and the TAMSD follows the same power law. The first passage time distribution has a power law decay for large times with a power $\alpha = 2 - H$ (Molchan, 1999). In addition, it is a Gaussian model, i.e., the distribution of the end-points of trajectories as well as all distributions of increments $x(t + \delta) - x(t)$ are Gaussian for all $t > 0$ and $\delta > 0$.

The results of section 3.3 suggest that $H = 0.9$. We can estimate the noise amplitude from the absolute values of the MSD in Fig. 7: the generalized diffusion coefficient is approximately 500 km$^2$/day$^{1.8}$ and 1000 km$^2$/day$^{1.8}$ for the southern and northern parcels, respectively. Note that a direct comparison of the generalized diffusion coefficient $k_H$ with the ones used in the literature for normal diffusion can not be performed, since their units of time and their role in the model are different. In our super-diffusive fBm model, diffusion generates some effective drift. Compared to a model of normal Brownian diffusion plus an explicit ballistic drift, our model without explicit drift term needs a larger generalized diffusion coefficient, since it also generates an effective drift.

Let our time discrete model of the time continuous fractional Brownian motion have a time resolution of 1 day, then the model noise amplitude is $\epsilon = \sqrt{2k_H}$. Hence, our model is given as $x(t) = \sum_{i=1}^{t} \epsilon \xi_i$, where $t$ is time in days and $\xi$ is power law correlated Gaussian noise with unit variance and zero mean, and $x(t)$ is the distance of the tracer from its starting position $x(0) = 0$ in zonal direction in km.

The results of the simulations are shown in Figs. 10 and 11. For the first passage time, we have approximated the distances in degrees by 111 km/degree, i.e., we numerically calculated first passage time distributions for distances of $x = 111, 222, 444,$ and 888 km.

Sanders and Ambjörnsson (2012) suggest that the first passage time distribution has the following probability density

$$\phi(t) \simeq c t^{H-2} \exp\left[-\gamma\left(\frac{s^2}{4k_H t^{2H}}\right)^\beta\right] , \tag{4}$$

where $k_H$ is the above mentioned generalized diffusion coefficient, $H$ is the Hurst exponent which was estimated to be $H \approx 0.9$ for the ocean drifters, and the three free parameters $c, \beta, \gamma$ to be fitted to the process. Their values can only depend on $H$ and $k_H$ and the units of space and time, but they have not been derived analytically for the case of fractional Brownian motion, so that we are free to fit them. But notice that they only affect the short term behaviour, since the exponential term tends to unity for large times $t$ and the probability decays with the power law $t^{H-2}$.

| $s$ | $c$ | $\beta$ | $\gamma$ | | $s$ | $c$ | $\beta$ | $\gamma$ |
|---|---|---|---|---|---|---|---|---|
| 1 | 0.25 | 1.9 | 0.04 | | 1 | 0.28 | 2.0 | 0.07 |
| 2 | 0.29 | 2.6 | 0.44 | | 2 | 0.35 | 2.5 | 1.36 |
| 4 | 0.36 | 4.0 | 81.43 | | 4 | 0.44 | 3.6 | $2.45{\cdot}10^{2}$ |
| 8 | 0.52 | 1.2 | 25.26 | | 8 | 0.59 | 4.0 | $1.02{\cdot}10^{5}$ |

**Table 1.** Fit parameters of the first passage time distribution Eq.(4) to the empirical first passage time of the tracers on the northern (left) and southern (right) hemisphere. Notice that the essential parameter $H$ has not been fitted but is given by the scaling of the MSD. Numerical fits by the non-linear least squares method.

.

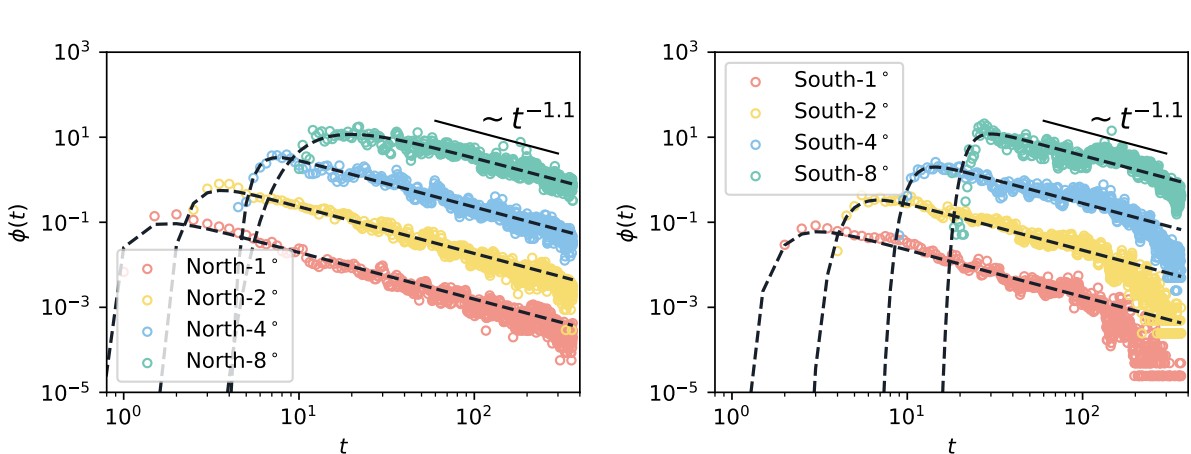

**Figure 10.** Comparison of the first-passage time distribution of the fractional Brownian motion model Eq.(4) with the empirical data of northern (left) and southern (right) subset of advected parcels. Note that for the better visual comparison of the data to the fBm model, their values are multiplied by a factor of 10 for $2°$ of 100 for $4°$ and of 1000 for $8°$.

We see in Fig. 10 that Eq. (4) with the parameters reported in Table 1 matches the empirical data from the oceanic tracers on the northern and southern hemisphere in an excellent way. Numerical simulations of this fractional Brownian motion process is in equally good agreement (Fig. 11), and we see that the fluctuations of the first passage time distribution of the oceanic tracer particles around the theoretical curve are of the same magnitude as the fluctuations of the numerical fBm trajectories, so that these can be explained by mere statistical fluctuations.

In Fig. 1 we see a quite clear westward drift for many of the trajectories. Such a drift term can be included also into the fractional Brownian motion model. A rough estimate of the drift velocity is about 5 km/day, so that it is slow compared to the diffusion. While this slow drift leaves the bulk of the first passage time distribution unaffected, it might lead to a cut-off at long times. Such cut-offs can be seen in data from the northern hemisphere, see Fig. 11: after the time of about 1 year, all tracers have reached the boundary, and the first passage time distribution drops to zero. Since this is a small correction, we do not study here the fBm-model with drift, also since the development of the corresponding mathematical theory is still incomplete.

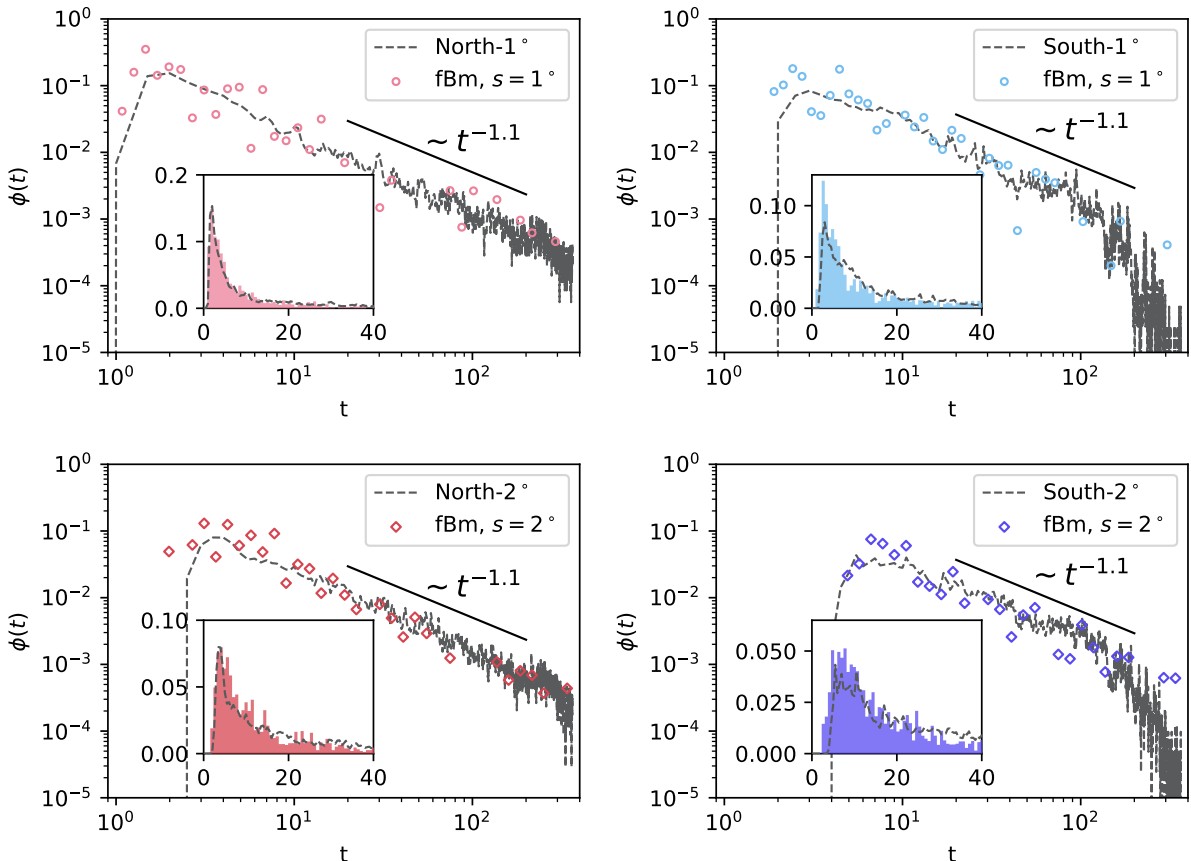

**Figure 11.** Comparison of the first-passage time distribution of the fBm simulation and the empirical data of northern (left, $k_H = 1000$ km$^2$/day$^{1.8}$) and southern (right, $k_H = 500$ km$^2$/day$^{1.8}$) subset of advected parcels. The short black solid lines show the asymptotic behaviour $t^{H-2}$ with the Hurst exponent $H = 0.9$ and the insets show the short time behaviour of the distribution on linear scale. The time step is chosen as $\Delta t = 0.0001$ for 6600 runs.

While a pure fBm-model has no preferred direction in space, i.e., particles would diffuse both westward and eastward with the same statistics, reality suggests here two modifications which can both be included in our model: there are boundary conditions (the American continent) which prevents particles from moving eastward (this might be taken into account by a reflecting boundary in our fBm-model), and there might be a westward drift of a few km/day which can be explicitly incorporated. Actually, the question "how much does diffusion contribute to the westward advection" is a very relevant one: our study tells us that an explicit drift term in our model, if really needed, should be quite low, in the range of 5 km/day or less. The strong persistence of the stochastic process, i.e., the long range temporal correlations of the random increments to the path, enforce that this changes their signs rarely, and hence produce over long time intervals motion which looks as if being directed. Therefore, in

the fBm-model, the apparent drift is a sole consequence of the persistence of fBm, and we are able to reproduce the statistical properties of the tracers without including a drift term.

## 4 Conclusions

We study oceanic transport in the equatorial Pacific by using observed and publicly available fields of the surface velocities. Within these time dependent fields, we study numerically the trajectories of tracer particles which are released on a latitude circle at $110^o$ west within the latitudes of $[-15^o,+15^o]$. We find non-trivial cross-correlations between advection indices and indices representing ENSO, with a time lag of 3 months. Beyond that, the erratic motion of individual particles show two phenomena: There is an overall westward drift, which has been described in many articles before. Our focus is on the random deviations from this mean flow.

We mention a possible limitation of the advection indices as defined by us. Since all the parcels are started from the eastern boundary of the Pacific basin, the tracer cloud spreads gradually westward, mostly, and covers different regions in different time intervals throughout a year. In this respect, the eastern basin seems to dominate the statistics. Firstly, we emphasize that the statistical theories of random advection processes refer to the same situation: a cloud of tracers released from a restricted region gradually affected by the changing flow field at an ever increasing time. Secondly, we argue that e.g., a uniformly distributed initial configuration (similar to drifting surface buoys in a given time instant) has also some shortcomings. Parcels starting from the western part of the basin approach quickly the coastal area of the numerous islands and the continent, therefore their trajectories are deflected by the local currents mostly in the meridional direction, and give spurious contribution to the westward drift statistics. This is also the reason why we restrict one experiment to one year.

By the statistics of the Mean Squared Displacement we see that the trajectory randomness is related to super-diffusion, i.e., the dispersion of particles is much faster than it would be the case for a normal diffusion process (Figs. 7 and 8). This might be of high relevance for the spreading of pollutants. A study of the first passage times shows us that this process can be well described by the fractional Brownian motion. This implies that the statistics is still Gaussian, but that the "noise" which makes the tracers diffuse has long range temporal correlations. Indeed, long range temporal correlations have been observed in many geophysical data sets before, such as temperature and precipitation time series, but here the Hurst exponent $H \approx 0.9$ is particularly large. So the stochastic process is strongly non-Markovian and has long memory.

We conclude that the collective effects of the spatial and temporal fluctuations of the velocity field which advects the particles has some self-similar structure which gives rise to a rather uniform time evolution on average over several years. In the end, our model contains only 2 parameters, the Hurst exponent $H$ and the generalised diffusion coefficient $k_H$. This is a huge simplification compared to the simulation of the advection process, and will be useful for the modelling and prediction of spreading of advected passive scalars such as pollutants, nutrients, or thermal energy in this region of the ocean.

*Code and data availability.* Global geostrophic velocity fields are openly available after registration at the E.U. Copernicus Marine Service

(https://resources.marine.copernicus.eu/products). Codes are based on standard Python modules described in Section 2 (Data and methods) in details. Advection experiments are performed by the package "Ocean Parcels" (Lange and van Sebille, 2017; Delandmeter and van Sebille, 2019) which is fully documented at https://oceanparcels.org/.

*Author contributions.* I.M.J. and H.K. designed research; I.M.J., A.P. and H.K. performed research; A.P. and H.K. contributed new numerical/analytical tools; I.M.J. A.P., J.A.C.G. and H.K. analyzed data; and I.M.J., A.P., J.A.C.G. and H.K. wrote the paper.

*Competing interests.* The authors declare no conflict of interest.

*Acknowledgements.* We thank the two anonymous Referees for the extremely careful reading and remarks which promoted significant improvements in the revised version. This work was supported by the Hungarian National Research, Development and Innovation Office under grant numbers FK-125024 and K-125171, and by the Max-Planck Institute for the Physics of Complex Systems. J.A.C.G. was supported by CNPq, Brazil, grant PQ-305305/2020-4.

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
