# Peer review of "Passive tracer advection in the equatorial Pacific region: statistics, correlations, and a model of fractional Brownian motion"

_Ocean Science, 2021_

## Author Comment (AC1)

**Response to Referee #1**

We are grateful for the comments of the referee. We appreciate very much the time and effort which the referee has devoted to our manuscript. The report will enable us to improve the presentation of the material and to put it into the right context. The referee has well summarised our paper in the first paragraph of RC1. Here we list all the critical remarks (in italics) and give an explanation of our points.

*RC1*: *With the application of a statistical model, the authors make an interesting effort to investigate the westward motion of water parcels in the equatorial Pacific from a exceptional viewpoint. However, I cannot see how the study as a whole contributes to an enhanced knowledge of the tropical Pacific. The results presented are largely known and what is new is not put into context with the literature.*

**Response**: It is absolutely true that the tropical Pacific has been in the focus of oceanographic research for a long time and it has a vast literature. We are somewhat confused, because the first sentence perfectly summarises the essence and novelty of our work, the second and third one question its relevance. Since this remark is further specified in the general and specific comments, please read our point by point responses below.

*General comments:*

*RC1*: *1. It is not entirely clear to me what the aim of this study is. The first part is largely a repetition of known results. The application of the statistical model in the second part leaves open the question what the benefit is. It is very important to make clear what the intention and also the significance of entire study is for the research community.*

*RC1*: *2. What is the benefit of using a fractional Brownian motion model in this context? Can this be applied elsewhere? What conditions have to be fulfilled for the mathematical model to be applied?*

**Ad 1+2**: We agree that we missed to explain what can be learned from the fact that the overall motion due to advection is a fractional Brownian diffusion process rather than ballistic transport. As it is already visible in our Fig. 1, the tracer trajectories perform quite irregular motion, even in years where a dominating westward drift is visible. At the end of a one-year-long path, quite a large number of particles is still close to where they have been released. These fluctuations, in particular their turns, slow down the particle motion. Their distances from the origin do not grow linearly in time as it would be for a pure drift. This "slowing down" can be seen in the mean squared displacement MSD, Fig. 8, which scales in time with a power of less than 2 (which would be the case for a pure drift), but with a power much larger than 1, as it would be

for classical diffusion process. This anomalous type of motion is called super-diffusion. The outstanding observation here is that the exponent $H$ of this anomalous diffusion process is a constant value for a large range of spatial and temporal scales, from 1-5000 km and from 2-365 days. This means that the collective effects of the spatial and temporal fluctuations of the velocity field which advects the particles has some self-similar structure which gives rise to a rather uniform time evolution *on average over several years*. In the end, our model contains only 2 parameters, the Hurst exponent $H$ and the generalised diffusion coefficient $k_H$. This is a huge simplification compared to the simulation of the advection process, and will be useful for the modelling and prediction of spreading of advected passive scalars such as pollutants, nutrients, or thermal energy in this region of the ocean.

On the other hand, as said, it is surprising to see that this anomalous diffusion process can be observed on such large ranges of spatial and temporal scales, where one might as well expect some cross-over effects, e.g., from (anomalous) diffusion on small scales to ballistic, drift dominated transport on larger scales. This shows that the irregularities of the velocity fields in space and time do not average out even on large scales and might reflect some kind of scale invariance of the turbulent motion of these hydrodynamic flows. This provides insight into the statistics of oceanic turbulence and might be of fundamental physical interest.

So we believe that there are two justifications for the relevance and the general interest in our findings.

Also, we have realised that when introducing our fBm model, we should explicitly say that an alternative model would be a model with an explicit ballistic drift term plus normal diffusion. The drift term would then describe the time dependence of the mean value of the longitudinal particle positions, and the normal diffusion term would explain why the variance of this distribution grows in time. Our analysis clearly shows that the statistical properties of such a quite plausible model are incompatible with the observations based on the numerical tracer advection.

***RC1****: 3. As vertical advection is not taken into account, I wonder what effect this would have on the results. How would a third dimension alter the fBm-model?*

**Ad 3**: We have no access to the vertical velocity field, and no model for the advection of particles in 3-d with the ocean surface as a boundary. Clearly, real transport is not restricted to the topmost horizontal layer of the ocean, but it is a useful approximation and also a standard approach to ignore the third dimension in such transport studies. Our fBm model is a model for the numerically generated tracer trajectories, and if we had 3-d simulations, we could test it for these as well. We assume that our numerical advected tracer paths are sufficiently realistic compared to, e.g., floaters released to the ocean, so that our fBm model should describe such experimental data (if they were available in sufficient amount) as well.

*RC1*: *4. As far as I understand, eventually the results show that the westward currents on the equator (SEC and NEC) include an advective term and a diffusive term (super diffusion). This alone is not new. What would be interesting is to have a quantitative measure of the percentage that diffusion contributes to the overall westward advection. Would this be feasible with the applied methodologies?*

**Ad 4**: Yes, while a pure fBm model has no preferred direction in space, i.e., particles would diffuse both westward and eastward with the same statistics, reality suggests here two modifications which can both be included in our model: There are boundary conditions (the American continent) which prevents particles from moving eastward (this can be taken into account by a reflecting boundary in our fBm model), and there might be a westward drift of some km per day which can be explicitly included in our model. Actually, the question "how much does diffusion contribute to the westward advection" is a very relevant one: Our study tells us that an explicit drift term in our model, if really needed, should be quite slow, in the range of 5km/day or less. The strong persistence of the stochastic process, i.e., the long range temporal correlations of the random increments to the path, enforce that these change their signs rarely, and hence produce over long time intervals motion which looks as if being directed. Hence, in the fBm model, the apparent drift is a sole consequence of the persistence of the fBm, and we are able to reproduce the statistical properties of the tracers without including a drift term into our fBm model. However, as far as the first passage time distribution is concerned, a slow drift (as mentioned, about 5km/day) helps to reproduce the shift of the maximum to the west. Without drift, our model's first passage time distribution agrees well with the one of the tracers up to 100 days. Beyond that, while also extending westward in a comparable way, it keeps its maximum closer to where the particles have been released. Also, an explicit westward drift term introduces a cut-off of the waiting time statistics as we observe it for the southern particles passing the 8 degree line.
So to summarise these lengthy explanations: In our model, the westward motion of the particles can be solely explained by the fractional diffusion process with a reflecting boundary in the east, but adding some slow ballistic drift of a few km per day gives small improvements in the reproduction of the observed first passage time distribution.

*RC1*: *5. The Northern and Southern advection indices show a largely coherent behaviour in all figures. What is the overall benefit of sub-dividing the trajectories into these two categories?*

**Ad 5**: We study northern and southern particles separately because *a priori* it is not clear whether these two advection processes have the same statistical properties. In Fig. 10, the first passage time distribution, we do see a cut-off due to a stronger westward drift for the southern particles.

We will include our explanations to items 1-5 in the revised version of the manuscript.

We are very grateful also for all of the specific comments. We will take all of them into consideration when revising the paper. Here we will respond only to those which contain a scientific question:

*Specific comments:*

**RC1***: Abstract: Some more information about the tracer experiments would be helpful for the reader, e.g., where are the particles released and what is the depth range of interest here.*

**AC1**: The tracers released from a meridional line at the longitude of 110°W, as explained in Section 2 (Data and methods) and illustrated in Fig. 1 by a black line. We will insert the requested information in the Abstract.

**RC1***: lines 60-62: Instead of repeating what is already stated in the figure caption, it would be nice to explain the difference between the two panels in Fig. 1, i.e., the effect of the weakened trade winds during El Nino impacting the surface currents on the equator.*

**AC1**: We fully agree that a large portion of inter-annual variability is due to ENSO. We spell this out in more detail in the revised version.

**RC1***: lines 89-95: This sounds very similar to the classical bootstrapping method. Is there a difference? Otherwise, mentioning the term known "bootstrapping" may help the reader better understand the explanation given in the text.*

**AC1**: The method of surrogate data is a modification of classical bootstrapping, where both the marginal distribution of the data and their serial correlations are conserved when creating the otherwise random samples. We will modify the wording of this section.

**RC1***: line 122-128: It seems too early to already talk about limitations of the study before even showing some results. E.g., the sentence "the eastern basin dominates in the statistics" does not really say anything without knowing of any specific numbers and figures. Also, in terms of the limitations mentioned and the comparison to drifting buoys, to me it seems like this really depends on the aim of the study. At this point, it is still not entirely clear to me what the aim of this study is. Is it to analyse the underlying physical context of the equatorial Pacific current system? Or else, is it to use this region as a test case to later apply the fBm-model?*

**AC1**: We shift these comments to a more suitable part of the paper.

***RC1****: Fig. 2: The coincidence between the two indices is remarkable and interesting. It is not really obvious from Fig. 1, where the meridional component is also clearly visible. I am wondering if it is in fact the difference in the mean of the indices that indicates the measure of pure zonal advection? However, by standardising the time series the mean is not considered anymore. For my personal understanding, I would be very interested in a figure with the non-standardised raw time series of AdI1 and AdI2. Apart from that, another question: Are parcels with an initial eastward component excluded from the analysis of the trajectories?*

[Figure]

ACfigure 1: Non-standardised advection indices $AdI_1$ and $AdI_2$ determined as ensemble averages of the monthly westward displacements. The first is based on the monthly zonal distance from the position at the end of the previous month, the second is the total distance travelled in a month (meridional components are also taken into account). **(a)** northern section; **(b)** southern section.

**AC1**: Please check ACfig. 1, where the advection indices are plotted without standardisation. The monthly mean values for both the total distance travelled and the zonal distance largely overlap indicating the fact that zonal drift dominates. Indeed, only tracers moving westward are incorporated in the statistics.

A further insight is provided by ACfig. 2, where the early phase of advection is illustrated for a few year. The typical advection path is directed either to west or east with small meridional deflections. Parcels travel to east are usually remained trapped close to the shoreline of the American continent.

[Figure]

ACfigure 2: The position of advected parcels after 20 days from start for a few years. The parcels are aligned along a continuous line, and their movement is typically toward west or east, with small meridional deflections.

**RC1**: *lines 135-136: Isn't this "decoupling" just a consequence of the shifting ITCZ which is located at 5°N in the mean? What would the indices look like if the NH and SH subsets were splitted at the ITCZ instead of the geographical equator? The coincidence would probably be much higher.*

**AC1**: Yes, the decoupling is the direct consequence of the presence of ITCZ. However, the instantaneous (daily) location of the ITCZ line is a delicate question, therefore ITCZ is usually illustrated by long time mean values in the literature. For this reason, we could not figure out an adequate method to separate parcels along the two sides of the ever changing ITCZ line.

**RC1**: *lines 137-140: The correlation plot also confirms that the seasons do not play a major role in the tropics, otherwise the 6-month lagged correlation would be the dominant peak.*

**AC1**: Seasons are indeed not important in the tropics considering the weather. However, the ITCZ position just mentioned above does have an annual cycle affecting the wind field and surface currents, thus the advection of parcels too.

**RC1**: *lines 140-141: Is there an expectation that they should follow the periodicity of one calendar year? If yes, what is the reason they do not? Please explain.*

**AC1**: As it is written in the text, an annual periodicity is expected because of the presence of the changing thermal equator. We cannot exactly explain why this signal is not present in the cross correlation plots, probably because we cannot locate exactly the ITCZ line and the statistics is somewhat distorted by this fact.

**RC1**: *Fig. 6: I do not think this figure is necessary as this is widely known and the according statement in the text (lines 146-147) does not need to be supported*

*by an extra figure.*

**AC1**: We do not think that figure 6 is widely known, where the confidence level is illustrated. Moreover this figure validates our code of determining cross-correlations between index values.

**RC1**: *lines 149-156: While this might be an interesting side note for an ENSO review paper, this entire paragraph outlining the historic evolution of the discovery of ENSO does not belong into a research paper.*

**AC1**: This short paragraph is not for experts, we would keep it for pedagogical reasons. We expect that some readers mostly interested in fBm will be happy to read a few lines about the background of ENSO.

**RC1**: *lines 158-163: I agree that the physics behind the time lag might not be trivial, however, it is quite well known that the subtropical cells (STCs) are very closely connected to interannual changes in equatorial Pacific SST, i.e. ENSO. The advection indices of this study are essentially representing the North and South Equatorial currents (NEC and SEC) which can be seen as a part of the overturning circulation. Various studies have dealt with the relationship between the different branches of the STCs and ENSO (e.g., Izumo 2005, Capotondi et al. 2005). Representatively citing Izumo 2005, the findings basically provide the following picture:*

*"The variations of all the branches of the cells are strongly correlated to ENSO (a slowdown during El Niño events and a strengthening during La Niña events). For an El Niño event (and conversely during a La Niña event), anomalous westerly zonal wind at 5°N (i.e., a decrease of the usual easterlies) cause a decrease in poleward Ekman meridional transport, leading chronologically to decreases in surface divergence, SEC, equatorial upwelling, and finally in EUC and in geostrophic meridional transports, thus in pycnocline convergence. These mass transport decreases strongly affect the heat balance and cause later an increase in SST of the eastern Pacific."*

*I am surprised that the authors find the SST to lead the advection indices which contradicts previous findings (e.g., as stated above). Might there be a sign error in the calculation? Or is this in fact an expression of the baroclinic adjustment through westward propagating Rossby waves?*

**AC1**: We thank for the two references. Both works are based on numerical simulations and deal with the rather complex, three dimensional equatorial current system of the Pacific. Specifically, Izumo focuses on the equatorial undercurrent (EUC) and less to the shallow meridional overturning cells which is the subject of the paper by Capotondi et al. Lag correlations are obtained for mean SST over the Nino3.5 area and the different branches of the current system.

We cannot see contradictions between our result and previous statement considering that mean SST (ONI) leads advection indices by a few months. The Aviso velocity fields are supposedly representative for the (trade) wind driven

surface currents and advection, while the above mentioned works focus on the interior flow properties such as equatorward mass convergence at the depth of the interior pycnocline. As Capotondi et al. formulated: "Increased (decreased) mass convergence is associated with increased (decreased) upwelling and lower (higher) SSTs. As a result, the two time series are largely anticorrelated (correlation coefficient is -0.91), but the largest correlation is found when the SST leads the meridional mass convergence by 2 months, a result that seems to be in disagreement with the view that changes in the strength of the STCs cause the SST changes."

Since we do not have information on the interior flow properties, we avoid pure speculations. We do not think that our result is related with westward propagating Rossby waves studied also by Capotondi et al., because "baroclinic Rossby wave propagation across the basin is a slow process (transit times can be of the order of a few years depending on the latitude), so that transport changes are not simultaneous at each longitude, but changes in the eastern part of the basin lead the changes in the western part."

*RC1*: *Fig. 8: I'm a bit confused by the x-label. Does the plot show MSD or time averaged MSD (TAMSD), as explained in the methods section? Also, I thought MSD already includes the ensemble mean, so what does EAMSD mean then (this also applies to Fig. 9)? Please explain.*

**AC1**: Many thanks for your careful reading. This is the MSD. Indeed, the notion EAMSD does not make sense, since MSD is already defined as an ensemble average. We have corrected the label.

*RC1*: *Fig. 10: The same colours should be used for the different longitudes in both panels. Also, there is no need to use different symbols.*

**Response**: We have updated the figure accordingly.

*RC1*: *line 205: The estimated diffusion coefficients of kH=5,800 m2/s (SH) and kH=11,600 m2/s (NH) are one to two orders of magnitude larger than typical diffusion coefficients used in OGCMs which range roughly from 10 m2/s to several hundreds m2/s. Do these results suggest that the horizontal diffusion is highly underestimated in models? Could the methodology presented here be used to better estimate diffusion coefficients globally that can then be applied to ocean models?*

**AC1**: A comparison of our generalized diffusion coefficient to ones used in the literature for normal diffusion can not be performed, since their units of time and their role in the model are different. In our super-diffusive fBm model, diffusion generates some effective drift, as discussed above. Compared to a model of normal Brownian diffusion plus an explicit ballistic drift, our model without explicit drift term needs a larger generalized diffusion coefficient, since it also generates the effective drift.

In any case, we are confident that it will be beneficial to blend the fBm model with other ocean models, if one can generalize the model from our study region to other regions.

*RC1: lines 226-227: How do the authors come to the conclusion that the drift is slower than the diffusion? Please explain.*

**AC1**: We are certain that the drift is slow compared to the diffusion since we studied fBm with additional drift terms of different magnitude. With drift terms which are of the order of 20 km/day which is what one observes when simply studying the mean of the particle distribution over time, we see clear differences in the statistics such as the MSD or the first passage time distribution.

*RC1: lines 237-238: Is the super-diffusive character not just a consequence of the underlying strong westward drift of the particles? Or has this drift been removed before computing the statistics (I did not notice anything like this in the text)?*

**AC1**: The question whether a numerically calculated MSD for ballistic motion can exhibit an exponent of 1.8 instead of 2 is well justified, since the error statistics of MSD calculations is still in its infancy (co-author H.K. has some recent work on this topic). No drift has been removed before performing the MSD and TAMSD analysis. Based on our experience with such type of analysis, we are sure that the observed sub-ballistic behaviour is real, since it covers more than 2 orders of magnitude in time. Also, we have repeated the very same analysis for numerically generated fBm trajectories and find very good agreement, while ballistic trajectories superimposed with white noise show a very clear $t^2$-scaling.

*RC1: Technical points:*
  *line 3: missing article: ...the equatorial Pacific...*
  *line 4: Typo: 20S*
  *line 21: Typo: ...flow fields...*
  *line 126: Typo: ...starting...*
  *line 174: Typo: remove one "the".*
  *line 212: "Actually" not needed.*
  *line 216: "Actually" not needed.*
  *line 232: Typo: ...Pacific...*

**AC1**: Thanks for spotting these infelicities, all corrected now.

---

## Author Comment (AC2)

**Response to Referee #2**

***RC2****: Analysis of tracer advection, particularly analysis of the Lagrangian trajectories, is of much importance. This paper investigate tracers' trajectories in terms of spatial pattern, relation to ENSO, statistical properties and the role of fractional Brownian motion. The topic is somewhat interesting, but some fundamental flaws, in terms of methodology and interpretation, are seen in this manuscript and important additional calculations or evaluations are needed.*

**AC2**: We are grateful for the detailed and thoughtful comments by the referee and the time and efforts the referee has devoted to our paper.

We fully understand and agree that in our presentation there is lack of discussion why the fractional Brownian motion model is useful. In the revised version we insert our related remarks. See also Author Comment 1 (Responses to Referee #1), since he/she formulated similar questions.

***RC2****: Major comments:*

*The key points of this study are: interannual variability of advection is related to ENSO, and well reproduce the statistical properties of the tracers' trajectories with a fractional Brownian motion model. It is well known that the oceanic advection is associated with ENSO cycle and there are actually numerous previous studies have examined the relationship between oceanic advection and ENSO in very details and on multi time scales. So, to me, the most important point of this paper the latter, i.e., the interpretation of the observed statistical properties of the tracers' trajectories using the fractional Brownian motion model. The authors examined the tracers' trajectories which are 1°, 2°, 4°, and 8° west of the starting position, claim that "the westward moving tracers can be mapped into a simple 1D stochastic process" and "numerical simulations of the fractional Brownian motion model that it is able to well reproduce the observed statistical properties of the tracers' trajectories". However, this may not correct. The advection of sea water particles is influenced by not only molecular-scale processes, but also small scale, mesoscale to large scale processes that related to geophysical fluid dynamics. Although the molecular-scale processes and even small scale processes may be stochastic, the geophysical fluid dynamical processes are not stochastic, whose major component is linear. One of the difference between the two kinds of processes is in the spatiotemporal scales. Because the spatial scale of mesoscale to large scale processes is much greater than 8 degrees and the time scale of large scale processes is longer than order of 100 days, the conclusions of the present paper is very misleading to the oceanography community. So, the authors may extend the range of tracers' trajectories in investigating the role of fractional Brownian motion in the observed trajectories.*

*The writing of this manuscript needs further improvement, including the logic structure and grammar.*

**AC2**:

To be specific, we neither claim nor believe that *"the advection of sea water*

*particles is influenced by . . . only molecular scale processes*". Our simulations of the motion of tracer particles are the solutions of deterministic equations of motion, where only the time and position dependent meso-scale surface velocity of the ocean currents enter. Hence our tracer particles are advected by this motion, and there is no diffusion on the molecular scales in our simulations. Due to the time step of the integrator (5 minutes), in each update step the particles make jumps of the order of 1 - 100 m. Our simulated tracers have a velocity of the order of 10 km/day, which is much faster than any molecular scale diffusion process.

However, the fluctuations of the velocity fields in time and space cause the tracers to perform quite irregular motion (Fig. 1). Despite some overall westward drift which is particularly strong in La Niña years, the motion is not ballistic. This means that the distance of a particle from its origin does not grow linearly in time. Our intention is to model these irregularities of the particles' motions in a statistical way. Our analysis shows that the particle motion represents an anomalous diffusion process. More specifically, fractional Brownian motion is the most suitable model which is well able to reproduce essential statistical properties of the spreading of the tracer particles, without the need to know details of the velocity fields. Instead, we have only two parameters, the Hurst exponent $H$ and the value of the generalised diffusion constant $k_H$, which are sufficient to reproduce the spreading of, e.g., pollutants on spatial scales from 10 to 10000 km and on temporal scales from 2-365 days. So our model is not a physical model for the molecular diffusion process, but a data driven effective description of the motion on meso-scales.

A more detailed comment on the temporal and spatial scales evidently is in order: We study the mean squared displacement of the particles up to distances of 10000 km, and on times up to one year (Fig. 8), where the fractional Brownian motion model reproduces well the observed anomalous diffusion on the whole range of scales. However, the nontrivial scaling of the MSD alone is not enough to identify fractional Brownian motion as the correct model among several other anomalous diffusion processes, so that we perform in addition the study of the first passage times. In this specific aspect, indeed, we restricted ourselves to distances of at most 900 km and 100 days. The reason for this is that in many years (one shown in Fig. 1), a large fraction of the particles does not cross the 8-degree-distance from the place where they have been released. This means that the number of particles which contribute to the first passage time distribution becomes the smaller the larger the distance, and already for 16 degrees we do not have any more a good statistical sampling of the probability density.

We will amend the revised manuscript along these lines, as well as we will, in response to referee 1, explain why such a stochastic model is useful and which insights we have gained by this approach.

We are most grateful also for the listing of many detailed comments, which we will all address in the revised version. Here we comment only on those which contain a scientific issue.

**RC2**: *Minor comments:*
*What does "advection strength" mean? The definition should be specified here*
*What kind of data or "numerical experiments" used? It's needed to replace*
*equatorial with "tropical" since the domain in this paper beyond the equatorial*
*region obviously.*

**AC2**: We will amend the abstract accordingly.

**RC2**: *Lines 50-51: Explain how did AVISO calculate geostrophic currents in*
*the equatorial region.*

**AC2**: Geostrophic balance does not hold at the equator, but the altimetry
data can still be used to infer velocities there, albeit with less confidence. The
AVISO data set implements the method of Lagerloef et al. (1999) between ±5°.
The basic balance underlying this method is the y-derivative of the meridional
geostrophic balance at the equator. The Lagerloef et al. method is essentially a
way of matching this regime with the geostrophic regime away from the equator.
The method has been validated with drifter data and has been demonstrated
to capture the major features and variability of the equatorial circulation.

**RC2**: *Figure 1: How did you choose these representative trajectories?*

**AC2**: If we had drawn the paths of all trajectories, they would superimpose
each other to some red blot. We therefore picked a sub-set of 20 of the initial
positions with regular spacing along the vertical line from 15°N to 15°S as those
where we draw the full path in red. If we had chosen another sub-set, the indi-
vidual paths would be different, but the qualitative behaviour of showing many
turns and deviations from the westward motion or even looping around would
be the same.

**RC2**: *Figures 3-6: (1) It seems that the 99 % confidence interval is shown in*
*plenty of grey lines? Why? (2) It should be specified in the captions what a*
*positive time lag means.*

**AC2**: As explained in the Section "Data and methods", confidence intervals
are estimated from a test set of 10,000 Fourier surrogate time series of the sec-
ond signal, and obtained the cross-correlations between the basic signal and
each surrogate. We dropped the lowest 50 and highest 50 values (defined by
the cross-correlation value at the maximum/minimum time lag of the original
result). The remaining 9900 cross-correlation functions are plotted individually
by grey lines.

**RC2**: *Section 3.2: I am confused what your purpose of section 3.2 is.*

**AC2**: As its title suggests, here we analyse the cross-correlations between the
ONI and SOI and the advection indices. The later is a global characterisation

of the advection intensity or advection strength.

***RC2***: *Figure 6: The relationship between ONI and SOI is well known. Figure 6 is not necessary and should be removed.*

**Ac2**: We did not find any similar cross-correlation plot with well defined 99 % confidence interval. Also this figure validate our code of computing the time lagged correlations.

***RC2***: *Line 174: what does "distance" mean?*

**AC2**: The "distance" is here the length in km of the projections of the starting point of a tracer trajectory (i.e., at 110°W) onto the equator and of the endpoint after time t. This includes the approximation that particles which move south or north of the equator have travelled in reality a shorter distance than those on the equator when their endpoints are on the same longitude, but this inaccuracy is small and cannot be well resolved, if the particles' starting and ending points are at different latitudes.

***RC2***: *Lines 176-177: The definition does not show an average. The authors may modulate the formula.*

**AC2**: We did it.

***RC2***: *Line 192: need to show the four positions in Fig.1*

**AC2**: We do not think that four meridional (vertical in this map projection) lines can help something. The longitudes and latitudes are clearly shown in the maps.

***RC2***:*Figure 7: why did you choose 1-8 degrees? How about longer distances?*

**AC2**: As explained above in the Response to the Major comments, for large distances the statistics breaks down, see ACfigure 1. Specifically, there are several years when the longitude of 150°W is crossed by only a few parcels, see also Fig. 1a in the manuscript. Note however, that while FPT statistics is sensitive to the dilution of tracers (the same number of parcels cover larger and larger area), the scaling behaviour holds for much larger distances and time intervals clearly illustrated by Figs. 8 and 9 in the manuscript.

***RC2***:*Lines 235-236: what are the "two phenomena"?*

**AC2**: The two phenomena are the mean drift, seen by the mean of the particle distribution, i.e., their average distance from the point where they were released, plus the random motion of the individual paths around this mean value.

[Figure]

ACfigure 1: Normalised first passage time distributions for large distances from the line of release at 110°W, see legends. **(a)** Southern section; **(b)** northern section.

So in summary, we see that the presentation of our material has to be improved, which we will do in the revised version. We are grateful to the referee for showing us where our article cannot be properly understood and where it was even misleading an expert reader. We hope that our responses will have helped putting our results into the right context.

We present a model which is valid on spatial scales up to several thousands of km and on time scales of up to 1 year. We believe that such a model which does not rely on the detailed knowledge of the velocity fields over space and time but only on two parameters, can be most useful to model and to forecast the spreading of advected tracers in the ocean. At the same time, it gives insights into the complex fluctuations of oceanic transport.

---

## Referee Report (RR1)

Review of the revised version of Janosi et al., "Passive tracer advection in the equatorial Pacific region: statistics, correlations, and a model of fractional Brownian motion", submitted to OS.

The authors have thoroughly responded to the majority of my comments by revising text passages and figures. There are a few comments that the authors have not reacted to (listed at the end of this document). Even if the authors do not see any need for modifications in the manuscript, I would be interested to receive their response. I have also listed some minor points that need further revision. After incorporating these minor revisions, I can suggest publication.

**Author's response to my original general comments 1 and 2:** Thank you for the clear explanation. This is exactly what I was missing in the first version of the manuscript.

**Author's response to my original general comment 5:** I understand that the analysis has been done for the Northern and Southern hemisphere separately, as the results were now known a priori. However, if your analysis reveals that there is no big difference between NH and SH, I do not think that for the publication it is necessary in each figure to show the same plot for NH and SH if they are the more or less same, i.e., if no distinct conclusions are drawn for a specific hemisphere. It would be sufficient to just mention in the text that this analysis has been repeated for both hemispheres and that the results are largely the same. If there is one aspect that clearly differs between the two hemispheres, it would be enough to show the respective plots just for this one figure.

**Author's response to my original specific comment on Figure 6:** Still not convinced that this figure is needed. Regarding the two positive peaks, I would not consider correlations below 0.2 physically meaningful, no matter if they are mathematically significant or not.

**Author's response to my original specific comment on lines 149-156:** I still recommend to remove this paragraph as it clearly draws the reader's attention away from the present study. There is an abundant amount of scientific and plain-language literature about ENSO and its historical evolution for interested readers.

**Original reviewer comments that have not been responded to (note that the line number corresponds to the first version of the manuscript):**
lines 72-73: It would be nice to motivate this sentence.

line 73: Using "ensemble mean" and "total mean" might be misleading here. Is there a sub-ensemble of all trajectories? I would assume the mean is in both cases taken over all trajectories. In this case, it would be clearer to refer to the "ensemble mean westward distance" and the "ensemble mean total trajectory length".

Eq.2: Not sure if this equation is really needed as it is widely known.

---

## Author Response (AR2)

Dr. Erik van Sebille
Topic Editor
Ocean Science

10/02/2022

Dear Dr. van Sebille,

We thank for your letter of decision and the response of Referee #1. We are happy that the overall opinion is positive, and you require minor revisions. Here we list the remaining points and the detailed answers, and the changes performed in the 2$^{nd}$ revised version.

**General comment 5:** We show only the separate results for NH and SH tracer subsets in subsection 3.2, where the cross-correlations with ONI and SOI is analyzed. We think that the cross-correlation function for NH and SH advection indices (Fig. 3) justifies a separated treatment, because of the statistically significant negative correlation at a time lag of 7 months. We think that a marked El Nino (or La Nina) episode has an effect on both side of the Equator, however in quite periods the two sections exhibit a kind of decoupling. In order to improve the explanation, we inserted the sentences below Fig. 3: "The observed relatively weak real-time positive correlation and the statistical feature of the phase shift by 7 months between the northern and southern tracer subsets suggested us to perform all subsequent statistical tests separately. We illustrate that the results are very similar, but not identical."

**General comment on Fig. 6:** We are convinced by now, thus we deleted the figure.

**Specific comment on lines 149-156:** We follow the recommendation and removed the paragraph.

**Line 72-73 (original numbers)**: In order to improve the definitions, we reformulated the sentences as follows: "Monthly advection indices $AdI_1$ and $AdI_2$ are defined by the ensemble mean values of two metrics: (1) zonal distance and (2) total trajectory length (meridional drift components are included) from the positions at the end of the previous month. We will see that there is no difference between the values of both standardized indices, demonstrating that westward drift absolutely dominates. On an absolute distance scale, $AdI_1$ is systematically lower than $AdI_2$, as expected, but the difference is negligible.

**Eq. (2)**: We would like to keep this equation to show the precise mathematical definition. According to our experience, the equation itself might be widely known, however the appropriate interpretation of the time lag at maximum (minimum) values can be problematic for many readers.

Sincerely Yours,

Imre M. Jánosi